# Biomimetic Incremental Domain Generalization with a Graph Network for Surgical Scene Understanding

**DOI:** 10.3390/biomimetics7020068

**Published:** 2022-05-28

**Authors:** Lalithkumar Seenivasan, Mobarakol Islam, Chi-Fai Ng, Chwee Ming Lim, Hongliang Ren

**Affiliations:** 1Department of Biomedical Engineering, National University of Singapore, Singapore 117583, Singapore; lalithkumar_s@u.nus.edu; 2Department of Computing, Imperial College London, London SW7 2AZ, UK; mobarakol@u.nus.edu; 3S.H. Ho Urology Centre, Prince of Wales Hospital, The Chinese University of Hong Kong, Hong Kong, China; ngcf@surgery.cuhk.edu.hk; 4Head & Neck Surgery, Singapore General Hospital, Singapore 169608, Singapore; lim.chwee.ming@singhealth.com.sg; 5Department of Electrical Engineering, The Chinese University of Hong Kong, Hong Kong, China; 6Shun Hing Institute of Advanced Engineering, The Chinese University of Hong Kong, Hong Kong, China

**Keywords:** surgical scene understanding, domain generalization, scene graph, curriculum learning

## Abstract

Surgical scene understanding is a key barrier for situation-aware robotic surgeries and the associated surgical training. With the presence of domain shifts and the inclusion of new instruments and tissues, learning domain generalization (DG) plays a pivotal role in expanding instrument–tissue interaction detection to new domains in robotic surgery. Mimicking the ability of humans to incrementally learn new skills without forgetting their old skills in a similar domain, we employ incremental DG on scene graphs to predict instrument–tissue interaction during robot-assisted surgery. To achieve incremental DG, incorporate incremental learning (IL) to accommodate new instruments and knowledge-distillation-based student–teacher learning to tackle domain shifts in the new domain. Additionally, we designed an enhanced curriculum by smoothing (E-CBS) based on Laplacian of Gaussian (LoG) and Gaussian kernels, and integrated it with the feature extraction network (FEN) and graph network to improve the instrument–tissue interaction performance. Furthermore, the FEN’s and graph network’s logits are normalized by temperature normalization (T-Norm), and its effect in model calibration was studied. Quantitative and qualitative analysis proved that our incrementally-domain generalized interaction detection model was able to adapt to the target domain (transoral robotic surgery) while retaining its performance in the source domain (nephrectomy surgery). Additionally, the graph model enhanced by E-CBS and T-Norm outperformed other state-of-the-art models, and the incremental DG technique performed better than the naive domain adaption and DG technique.

## 1. Introduction

Surgical scene understanding paves the way for embedding situational awareness into robotic systems assisting surgeons in minimally invasive surgery. The current systems’ inability to understand the surgical scene renders it difficult to address the key limitations of minimally invasive surgery—lack of haptic sense, situation responsiveness, and lack of a standard surgeon evaluation matrix [1,2,3,4,5]. When given the ability to mimic human scene understanding, such as instrument–tissue interaction detection, the system could trigger haptic feedback based on tissue-instrument interaction, serve as a second pair of eyes monitoring the surgery, evaluate the surgeon’s performance, and generate surgery reports. In recent times, many deep learning models that fundamentally perform scene understanding have been introduced for surgical workflow detection [6], surgical phase detection [7] and surgical skill evaluation [8,9]. As the instrument–tissue interaction model is aimed at capturing key interactions happening in surgery, the detected sequence of key interactions could potentially improve the surgical workflow recognition and surgical skill evaluation tasks. By detecting key interactions, it can also be potentially integrated with teaching by demonstration modules, to teach and evaluate surgical residents. While instrument–tissue interaction detection models have the potential to significantly improve such downstream tasks, realizing a interaction detection model is challenging. With the number of instruments and interactions constantly changing during a surgery, every scene (frame) may have different numbers of inputs and outputs (different numbers of instruments and interactions), and thereby each scene poses a unique Euclidean space problem. Furthermore, each interaction between the tissue and an instrument in a scene could depend on various local and non-local factors: (a) tissue features, (b) primary instrument features, (c) secondary instrument features (when the secondary instrument is used to manipulate the primary instrument), (d) instrument positions, and (e) overall surgical scene features. These factors present the instrument–tissue interaction detection task as a non-Euclidean space problem, making it challenging to solve it using traditional convolutional neural networks [10]. While these limitations have been addressed by employing graph networks [5], the task gets further complicated when the model needs to be extended to new surgical domains. Firstly, the domain shifts (the changes in instruments and tissue features) in new surgical scenes raise concerns over any model’s performance. Secondly, the inclusion of novel instruments and tissues in the new surgical domain warrants the model being retrained on both the old and new datasets.

The different numbers of inputs and outputs in each scene, domain shifts, continued inclusion of novel instruments and tissues in the new surgical procedures, and scarce data availability make the interaction detection model domain-specific and less effective for general minimally invasive surgery. A naive solution would be to train a model specific to each surgical procedure. However, common surgical tools used in various surgeries may play different roles based on the surgery. With the lack of massive procedure-specific datasets, it is better to create a domain-generalized model encoded with the universal purpose of each tool that can output its interaction with the tissue based on the surgical scene. To address this, with the assumption of no novel instrument or interaction in the new domain, the old network could be domain adapted by further training only on the new domain dataset. Alternatively, the network could be retrained on both the old and new domain datasets by scarifying the training time. While both methods could serve as an interim solution, the continued emergence of novel instruments and new surgical procedures makes these solutions inefficient eventually. Therefore, there is a dire need for a method to domain-generalize the surgical scene understanding model to include new domain shifts, instruments, and tissues from new surgical procedures without the need to retrain the model from scratch.

Humans with a specific set of skills often display the ability to incrementally learn a new skill for a similar domain with ease, without unlearning the previous skills. For instance, cricket players can quickly learn and adapt to playing baseball without scarifying their skills in cricket. Furthermore, to maintain and improve their skills in both domains, they can train at regular intervals in both domains instead of re-learning those skills from scratch. Mimicking this, we propose an incremental domain generalization technique, where a graph model initially trained for instrument–tissue interaction detection in nephrectomy surgical scenes is incrementally domain generalized to detect interactions in transoral robotic surgical scenes. Firstly, inspired by the use of visual, semantic, and word2vec features in a graph network for the task of human–object interaction detection, we employed the visual-semantic graph attention network (VS-GAT) [11] model to perform surgical scene understanding with a nephrectomy surgery dataset. Secondly, to incrementally domain generalize the graph model, we incorporated the IL technique [12] and the knowledge distillation technique to extend the model to the target domain. The IL technique was employed to include novel instruments in the target domain, and the knowledge distillation technique was employed to generalize the model to domain shift in the target domain without adversely scarifying the performance in the source domain. Additionally, the graph network’s fundamental performance (without links to domain generalization) is enhanced by incorporating T-Norm and our proposed E-CBS. Our key contributions in this work are as follows:–Incremental DG: Integrating (i) a graph network with knowledge distillation and (ii) FEN with IL, to domain generalize the instrument–tissue interaction detection model. The proposed model was first trained for nephrectomy surgery and then incrementally domain generalized to transoral robotic surgery.–Enhancing a graph model for surgical scene understanding: The graph network and FEN were enhanced by T-Norm and our proposed E-CBS. The E-CBS introduced (i) 2D convolution layers with its kernel (3 × 3) weights based on LoG and Gaussian to enhance the FEN and (ii) a 1D convolution layer with kernel (1 × 3) weights based on LoG to enhance the graph model. T-Norm normalizes the network logits during training and inferences to improve model reliability.–Quantitatively and qualitatively, we proved that (i) our model enhanced by E-CBS and T-Norm outperformed other state-of-the-art models, and (ii) our knowledge-distillation-based incremental DG performed better than the naive domain adaption and DG technique.

## 2. Methods

To perform ***surgical scene understanding***, we employed a graph network inspired from VS-GAT [11] that uses ResNet18 [13] as the FEN. Firstly, the performance of the graph network in interaction detection was enhanced using E-CBS and T-Norm. Secondly, to achieve ***incremental DG in surgical scene understanding***, a two-tier approach was adopted: (a) As the graph network relies heavily on its embedded features, the FEN was extended to the target domain using IL [12] to cater to the domain shifts and the novel instruments. (b) Motivated by the IL technique [12], we then adopted the teacher–student training regime from knowledge distillation to extend the graph network to a new domain dataset without sacrificing much of its performance on the old dataset. The graph network was first trained on the source domain. Considering this trained model as a teacher network, a copy (student network) was then generalized to the target domain.

### 2.1. Surgical Scene Understanding

***Feature extraction network (FEN):*** ResNet18 [13] is initially trained to classify the instruments and tissues in the dataset before being employed as the FEN. This enables its penultimate layer to perform optimal feature extraction. The FEN is further enhanced by appending E-CBS and incorporating T-Norm to the output logits. ***Graph network:*** The surgical scene understanding task is theorized as G(V,E,Y), where G() is the graph network, V and E are the nodes and edges embedded with the features extracted using the FEN, and Y is the detected interaction. Given a surgical scene with bounding boxes, it is inferred as a sparse graph $G_{g}\in G$ to detect the interactions $G_{y}$. As shown in Figure 1, the visual features ($F_{vi}=$ {F,M,S}) of the instruments and tissues are embedded into the visual graph $G_{v}$ nodes. A pre-trained model of word2vec [14] is employed to embed nodes of the semantic graph $G_{s}$ with the node name. The nodes of $G_{v}$ and $G_{s}$ are then propagated ($G_{v}^{\prime }$ and $G_{s}^{\prime }$) and combined into a single graph $G_{c}$. Finally, The edges of the $G_{c}$ are embedded using the spatial features $F_{sp}$ [11]. Upon aggregation in $G_{c}$, the readout function utilizes the edge features ($G_{c}\left(E\right)$) to predict the interaction. We further enhanced the graph network by appending the proposed E-CBS to the visual graph’s node aggregation and T-Norm to the end of the graph’s readout function.

#### 2.1.1. Enhanced-Curriculum by Smoothing

CBS [15] proposes the use of 2D Gaussian kernels as anti-aliasing filters to augment the input features. This allows the model to progressively learn better feature representations. During training, the σ value of the Gaussian kernel is reduced by a decay rate every few epochs, to gradually allow more high-frequency features to pass through the model. Here, we introduce the LoG kernel instead of the Gaussian kernel to progressively allow high-frequency features and increase attention to highly intensity-varying regions. Given (*x*, *y*), the pixel locations in the kernel, the weights of the LoG kernel are calculated based on:(1)$$LoG\left(x,y\right)=-\frac{1}{\pi \sigma^{4}}\left[1-\frac{x^{2}+y^{2}}{2\sigma^{2}}\right]e^{-\frac{x^{2}+y^{2}}{2\sigma^{2}}}$$

***FEN:*** E-CBS is employed in the FEN using both the LoG and Gaussian kernels. In ResNet18 [13], an LoG-kernel-based 2D convolution layer is appended to the initial convolution layer and a Gaussian-kernel-based 2D convolution layer is added to every residual block. ***Graph network:*** E-CBS is employed using a LoG-kernel-based 1D convolution layer at the visual graph ($G_{v}$)’s edge function. It is aimed at smoothing the features aggregated during the node propagation at the initial training stages.

#### 2.1.2. T-Norm

The instruments and interaction distribution across both the source and target domains represent a long-tail problem, common in scene understanding and medical datasets. We normalize the temperature by scaling the logits of the FEN and the graph network (Equations (Equation 2) and (Equation 3)) during training and inference and study its effect on model calibration.

### 2.2. Incremental DG in Surgical Scene Understanding

***FEN:*** Initially, the FEN was trained to classify 9 classes (1 tissue and 8 instruments) found in the source domain. We then implemented IL [12] to extend the source-domain-trained FEN to classify 11 classes. It helped include two novel instruments found in the target domain and adapt to the domain shifts. This allowed FEN to perform optimal feature extraction in both domains. ***Graph network:*** To achieve DG, two key problems, (i) new instruments and (ii) domain shift, must be addressed. As each scene frame is considered as a sparse graph $G_{g}$ and the graph network is robust in handling various numbers of nodes without a fixed sequence, new instruments can be added by further training the graph network on the target domain’s scene without any change to the network structure. Further training on the target domain also allows the network to adapt to the domain shifts. In the proposed DG method, the network is initially trained on the source domain based on the multi-label soft margin loss ($L_{MSL}$) between the network output and ground truth. The source-domain-trained network is then domain generalized based on knowledge distillation, inspired by the works in IL [12]. The source-domain-trained network is considered a teacher network. A copy of it, the student network, is then trained on (i) the full target domain dataset based on $L_{MSL}$ between the network output and ground truth; and (ii) a sample (*n*) of the source domain dataset based on $L_{MSL}$ between the network output and ground truth, and knowledge distillation loss ($L_{KD}$) between the teacher and student network logits. The $L_{KD}$ enables the student model to retain on the source domain while generalizing to the target domain. The DG loss is given by: $L_{DG}=L_{TD}+L_{nSD}$, where:(2)$$L_{TD}=L_{MSL}\left(\frac{O_{SM}}{T_{Norm}},GT\right)$$
(3)$$L_{nSD}=L_{MSL}\left(\frac{O_{SM}}{T_{Norm}},GT\right)+0.5*L_{KD}\left(\frac{O_{SM}}{T_{Norm}},O_{TM}\right)$$

$L_{TD}$ is the target domain loss, $L_{nSD}$ is the sampled source domain loss, $O_{SM}$ is the student model output, $O_{TM}$ is the teacher model output, and $L_{KD}$ is calculated using cross-entropy loss. Finally, the student model is fine-tuned to both domains by training on equally sampled (*n*) source and target domain.

## 3. Experiment

### 3.1. Dataset

***(a) Incremental DG in surgical scene understanding:*** Instrument Segmentation Challenge 2018 [16] consists of robotic nephrectomy procedure video frames and was utilized as the source domain dataset. Our own SGH TORS 2020 consists of transoral robotic surgery video frames and was utilized as the target domain. As the training set, the source domain consisted of 1560 images (11 sequences: 2–4, 6–7, 9–12, 14–15) and the target domain consisted of 143 images from the target domain (15 sequences: 1–15), each image of dimensions 1280 × 1024 pixels. In the test set, the source domain included 447 (3 sequences: 1, 5, 16) and the target domain comprised 122 images (7 sequences: 16–22). In the source domain, the 8th sequence was not made publicly available and 13th sequence lacked instrument–tissue interactions. The instrument–tissue interactions were annotated by our clinical experts. In total, 13 interactions were identified across the source and target domains—idle, grasping, retraction, tissue manipulation, tool manipulation, cutting, cauterization, suction, looping, suturing, clipping, staple, and ultrasound sensing.

***(b) FEN:*** Both source and target domains were exploited to train the FEN to classify tissues and instruments. Tissues (kidney/tissue) and 10 instruments (bipolar forceps, prograsp forceps, large needle driver, monopolar curved scissors, ultrasound probe, suction, clip applier, stapler, Maryland dissector, and spatulated monopolar cautery) were cropped from the datasets and resized to 224 × 224 pixels. In total, the training set consisted of 7019 images and the test set consisted of 1460 images.

***(c) E-CBS:*** Cifar10 [17] was employed to benchmark the performance of the proposed E-CBS as an independent study. The cifar10 [17] comprises 10 classes, with 50,000 images (32 × 32) in the training set and 10,000 images in the test set.

### 3.2. Implementation Details

**(a) Graph network:** The graph network was trained on each domain using a constant learning rate = $10^{-5}$, batch size = 32, epoch = 250, and adam optimizer. In the graph network, the T-Norm was set to 1.5, and the E-CBS was initialized with σ=1.0 with a decay of 0.985 every 20 epochs. During DG, a sampling size *n* = 143 was used to sample frames from the source domain. For the final balanced fine-tuning phase during DG, learning rate = $10^{-6}$, epoch = 80, and n=143 were adopted.

**(b) FEN:** All variants of FENs, with and without label-smoothing [18], with and without IL, with and without E-CBS, and with and without T-Norm were trained using cross-entropy loss and a stochastic gradient descent optimizer, and with the same hyper-parameters: decaying learning rate starting at $10^{-3}$, epoch = 30, and batch size = 20. For IL, memory size = 50 and fine-tune epoch = 15 were adopted. The FE network used a T-Norm = 3.0, and its E-CBS was initialized with a sigma=1.0 with a decay factor of 0.9 for every 5 epochs.

**(c) E-CBS:** Three variants, (i) the CBS [15], (ii) the CBS (LoG): where all Gaussian kernels are replaced by LoG kernels, and (iii) the E-CBS, were experimented on with ResNet18 [13]. Each variant was benchmarked under three criteria: (a) σ = 1.0 and decay = 0.9, (b) σ = 1.0 and decay = 1.0, and (c) σ = 2.0 and decay = 0.9. To prove it is global application, all variants were trained on the Cifar10 [17] dataset, using cross-entropy loss, a stochastic gradient descent optimizer, and hyper parameters: learning rate = $10^{-2}$, batch size = 64, and epoch = 200.

All code (https://github.com/lalithjets/Domain-Generalization-for-Surgical-Scene-Graph, accessed on 12 April 2022) in this work was implemented using the PyTorch framework on three GPUs (1× Nvidia GTX 1080 Ti and 2× Nvidia GTX Titan X).

## 4. Results

Firstly, the performance of our proposed model trained only on the source domain under the unsupervised DG technique is benchmarked against other state-of-the-art models and the base model (VS-GAT) in Table 1. Our model outperformed other models in the source domain in terms of and target domains in terms of accuracy (Acc), mean average precision (mAP), and recall. In addition, our model (enhanced by E-CBS and T-Norm) outperformed the base model in both the source and target domains. This proves that using the T-Norm and the proposed E-CBS improves the model performance. While GPNN [19] and Islam et al. [5] models outperformed our model in terms of mAP, due to our model’s high performance in terms of Acc and its inclusion of word-text embedding (which could help in the future downstream captioning/visual question and answering task), our model is further studied.

Secondly, the performance of our model trained using our proposed knowledge distillation-based incremental DG technique is also compared against our model trained using the naive domain adaption and DG techniques (Table 2). Under naive domain adaptation, the model was initially trained in the source domain in phase 1. The model was then domain adapted by training only on the target domain in phase 2. Under naive domain generalization, upon training only on the target domain in phase 2, the model was fine-tuned on *n* samples from both domains. It is observed that our proposed incremental DG technique improved the model’s performance in both the source domain and target domain, extending the model to the target domain while retaining its performance in the source domain. Figure 2 reports the qualitative performance of our incremental DG model in detecting the interaction between the tissue and instruments. Despite a significant domain shift in the target domain and the presence of two novel instruments, the model managed to detect instrument–tissue interaction without a significant sacrifice in its performance in the source domain.

Thirdly, an ablation study of our model trained using unsupervised DG (trained only on source domain) technique and using our incremental DG technique, with and without label-smoothing [18], with and without E-CBS, and with and without T-Norm, is also reported in Table 3. Table 3 shows that the FEN trained using the IL [12] improves model performance in both the source domain (Acc) and target domain (Acc and mAP). The proposed E-CBS mostly improved the model’s Acc, mAP, and recall in both unsupervised DG and DG. It was also observed that our proposed model outperformed/performed on par with other best variants in both unsupervised DG and DG. Medical datasets usually pose a long-tail problem that often leads to model miscalibration. Therefore, in addition to Acc, mAP, and recall, the model’s performance is also evaluated based on the reliability diagram [22,23] in Figure 3. It is observed that the T-Norm reduced the deviation between the model’s performance and the diagonal line, improving the model’s reliability.

Fourthly, an ablation study was conducted on the FEN with and without IL [12], E-CBS, and T-norm, in classifying the tissues and instruments found in the source and target domains (Table 4). It was observed that, in both cases, (i) without IL in classifying nine classes and (ii) with IL in classifying 11 classes, E-CBS and T-Norm were observed to quantitatively increase the FEN’s performance.

Finally, an ablation study is reported in Table 5 that shows the ResNet18 [13] model’s performance on the Cifar10 dataset when combined with (i) CBS [15], and (ii) our proposed CBS (LoG): all Gaussian kernals replaced by LoG kernels and our final E-CBS. Table 5 shows that both CBS (LoG) and E-CBS quantitatively outperformed CBS [15] in most cases. In particular, E-CBS outperformed the CBS [15] in all test cases. CBS [15] employs Gaussian kernels to blur and limit the amount of features that propagate through the model and allow the model to learn progressively. As LoG gives attention to edges, replacing the Gaussian at every stage (image size reduces as it passes through different convolution layers), edges detected at initial stages (higher scale) could be dropped as the scale reduced. This could result in the loss of key features. Enforcing edge attention at multiple scales could have also induced noise. Having an LoG kernel at the initial layer to progressively reduced attention to edges and then using Gaussian kernels to progressively allow those edge features and surface features to pass through the network seems to offer a better solution.

## 5. Conclusions

Mimicking a human-like incremental learning regime, we proposed an incremental domain generalization technique to generalize an instrument–tissue interaction detection graph model to a target domain without sacrificing its performance on the source domain. This is achieved by incorporating an enhanced graph network with a teacher–student model-based knowledge distillation to handle domain shifts and an enhanced feature extractor network with incremental learning to include novel instruments. Furthermore, we introduced an E-CBS that uses (i) 2D convolution layers based on the Laplace of a Gaussian kernel and a Gaussian kernel to enhance the performance of feature extractor, and (ii) a 1D convolution layer based on the Laplace of a Gaussian kernel to enhance the graph network. We also employ temperature normalization to both the feature extraction and graph network’s output logits during training and inference to improve model calibration. Incrementally, domain-generalized graph models can expand surgical scene understanding across various surgical procedures with novel instruments and domain shifts. Our proposed model was able to detect interaction with an inference time of 52.47 ms, making it suitable for real-time applications. While our solution offers better domain generalization results compared to other existing methods, its performance on the target domain is still limited and needs further improvement for real-world applications. Following our current single–teacher single-student approach, a multiple-teacher (one for each domain) single-student domain generalization approach could be explored to further improve performance on both source and target domains.

## Figures and Tables

**Figure 1 biomimetics-07-00068-f001:**
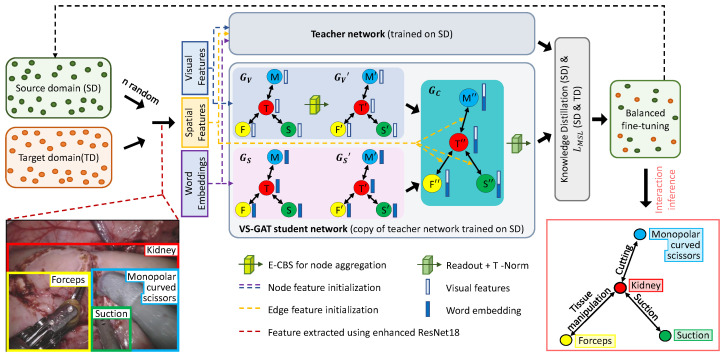
**Graph network for surgical scene understanding:** Given a surgical scene with bounding boxes, visual features of tissues and instruments are extracted using the FEN. Visual features are embedded in the visual graph ($G_{v}$); nodes and node names are embedded in the semantic graph ($G_{s}$) nodes. To improve graph network performance, E-CBS is appended to $G_{v}$’s node aggregation. Upon node aggregation in both graphs, they are combined to form $G_{c}$, having edges embedded with spatial features. Upon $G_{c}$ aggregation, the readout function processes the edge features to predict interaction logits which are then temperature normalized. **Incremental DG:** Given a model trained on source domain, DG is achieved in 2 tiers. Firstly, the FEN is naively trained using IL [12] to include novel instruments and domain shifts. Secondly, the graph network is domain generalized based on the student–teacher training regime in knowledge distillation. A graph network initially trained in the source domain is considered the teacher model. A copy is then taken as a student model and further trained on the target domain and random samples from the source domain. Finally, the student model is fine-tuned on a balanced distribution from the source and target domains.

**Figure 2 biomimetics-07-00068-f002:**
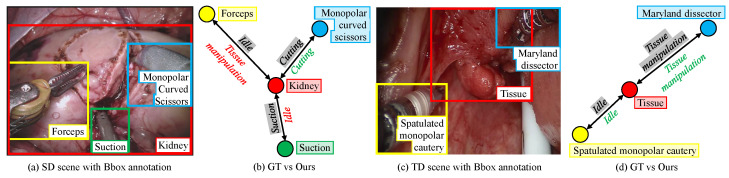
Qualitative analysis: (**a**) Source domain surgical scene with annotated bounding box (Bbox). (**b**) Ground truth (GT) interaction vs. our model’s prediction (in red and green text). (**c**) Target domain surgical scene with annotated Bbox and (**d**) GT interaction vs. our model’s prediction.

**Figure 3 biomimetics-07-00068-f003:**
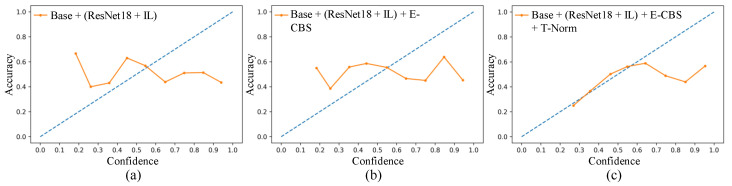
Reliability diagram: (**a**) Base + (ResNet18 [13] + IL [12]), (**b**) Base + (ResNet18 [13] + IL [12]) + E-CBS and (**c**) Base + (ResNet18 [13] + IL [12]) + E-CBS + T-Norm.

**Table 1 biomimetics-07-00068-t001:** Comparison of our proposed model’s performance in the source and target domains against the state-of-the-art scene graph models when trained solely on source domain.

Graph Network	Feature Extractor	Source Domain			Target Domain		
	ResNet18 [13]	Acc ↑	mAP ↑	Recall ↑	Acc ↑	mAP ↑	Recall ↑
GAT [20]	Vanilla	33.21	0.0973	-	33.25	0.0773	-
G-Hpooling [21]	Vanilla	33.21	0.1523	-	33.25	0.0790	-
GPNN [19]	Vanilla	55.00	0.1934	-	29.52	0.1980	-
Islam et al. [5]	Label-smoothing [18]	48.02	0.2157	-	29.52	**0.1947**	-
VS-GAT [11]	Vanilla	62.96	0.2682	0.2888	35.49	0.0999	**0.1327**
Ours (VS-GAT [11] + E-CBS + T-Norm)	IL [12] + E-CBS + T-Norm	**63.31**	**0.2975**	**0.2988**	**39.25**	0.1009	0.1268

**Table 2 biomimetics-07-00068-t002:** Comparison of the model performance trained using our proposed knowledge distillation (KD)-based incremental DG technique against the performances of models trained using naive domain adaptation and DG techniques on the source domain (SD) and target domain (TD).

Technique	Phase 1	Phase 2		Fine-Tunning	Source Domain			Target Domain		
	SD	TD	KD (*n*SD)	*n*SD + *n*TD	Acc ↑	mAP ↑	Recall ↑	Acc ↑	mAP ↑	Recall ↑
Domain adaptation	✓	✓	✕	✕	42.89	0.3122	0.1994	32.76	0.1211	0.1489
DG	✓	✓	✕	✓	44.10	0.3273	**0.2189**	32.42	0.1185	**0.1694**
Ours (incremental DG)	✓	✓	✓	✓	**56.59**	**0.3339**	0.2138	**33.11**	**0.1407**	0.1515

**Table 3 biomimetics-07-00068-t003:** Ablation study of our proposed model trained using unsupervised DG and incremental DG in instrument–tissue interaction detection in the source and target domains.

Model	Feature Extractor (ResNet18 [13])		E-CBS	T-Norm	Source Domain			Target Domain		
	Label-Smoothing [18]	IL [12]			Acc ↑	mAP ↑	Recall ↑	Acc ↑	mAP ↑	Recall ↑
**Unsupervised DG**										
Base	✕	✕	✕	✕	62.96	0.2682	0.2888	35.49	0.0999	0.1327
Base	✓	✕	✕	✕	63.82	0.2649	0.2922	35.15	0.0988	0.1171
Base	✕	✓	✕	✕	63.57	0.2673	0.2650	36.86	**0.1012**	0.1223
Base	✕	✓	✓	✕	63.65	**0.3129**	0.2986	**41.30**	**0.1012**	**0.1467**
Base	✕	✓	✕	✓	**64.51**	0.2594	0.2987	35.84	0.0965	0.1223
Ours	✕	✓	✓	✓	63.31	0.2975	**0.2988**	39.25	0.1009	0.1268
**Incremental DG**										
Base	✕	✓	✕	✕	55.47	0.3072	0.2025	**35.84**	0.1178	**0.1949**
Base	✕	✓	✓	✕	**57.71**	0.2869	0.2088	33.11	**0.1502**	0.1876
Base	✕	✓	✕	✓	54.87	0.3123	0.2000	34.47	0.1070	0.1877
Ours	✕	✓	✓	✓	56.59	**0.3339**	**0.2138**	33.11	0.1407	0.1515

**Table 4 biomimetics-07-00068-t004:** Ablation study of the FEN trained in classifying the tissues and instruments found in the source and target domains.

Model				Acc ↑	
ResNet18 [13]	IL [12]	E-CBS	T-Norm	Source Domain (9 Classes)	Source and Target Domain (11 Classes)
✓	✕	✕	✕	35.24	-
✓	✕	✕	✓	35.24	-
✓	✕	✓	✕	**39.21**	-
✓	✕	✓	✓	**39.21**	-
✓	✓	✕	✕	-	31.85
✓	✓	✕	✓	-	28.90
✓	✓	✓	✕	-	32.19
✓	✓	✓	✓	-	**33.49**

**Table 5 biomimetics-07-00068-t005:** Comparison of the proposed E-CBS and CBS (LoG) against the CBS [15].

Modules and Parameters					Acc ↑
σ	Decay	Model	Initial Conv Layer	Residual Blocks	
		CBS [15]	Gaussian	Gaussian	89.23
1.0	0.9	CBS (LoG)	LoG	LoG	88.17
		E-CBS	LoG	Gaussian	**90.21**
		CBS [15]	Gaussian	Gaussian	84.70
1.0	1.0	CBS (LoG)	LoG	LoG	**88.35**
		E-CBS	LoG	Gaussian	87.02
		CBS [15]	Gaussian	Gaussian	86.48
2.0	0.9	CBS (LoG)	LoG	LoG	**88.23**
		E-CBS	LoG	Gaussian	87.01

## Data Availability

The public datasets used in this work are mentioned in Section 3.1. The introduced target domain dataset will be made public upon paper acceptance.

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
