# Peer review of "Biomimetic Incremental Domain Generalization with a Graph Network for Surgical Scene Understanding"

_biomimetics, 2022, doi:10.3390/biomimetics7020068_

Round 1

Reviewer 1 Report

This work tackles the problem of surgical scene understanding. Specifically, it focuses on incremental domain generalization in the task of instrument-tissue interaction. Their proposed model for Domain Adaptation is based on a teacher-student network and combined with a graph model and a Visual Feature Extraction model based on resent-18. The Graph model consists of a Visual and Semantic Graph part that are then combined together.

To this end, they employ the incremental learning paradigm, where new instruments are learned without forgetting the old ones. Furthermore, they propose Laplacian of Gaussian based curriculum smoothing, and temperature normalization to improve the generalization performance. 

Strengths:

  • The task of Domain Generalization is very well motivated in the medical domain due to the many different potential procedures and challenges but the strongly limited amount of available data.
  • Employing incremental learning to deal with two different surgical datasets, despite the addition of new instruments.
  • The authors introduce many different components to their methods but still provide ablation studies regarding all of them.
  • Proposition of the E-CBS, to progressively allow higher frequencies. 
  • Authors present an exhaustive an ablation study.
  • They make their code publicly available.

Weaknesses:

  • It is not clear if the main goal is to keep the performance on the original source domain or adapt to the target domain. This makes comparing different approaches challenging.
  • Target Domain Performance: Comparing results from Table 1 and Table 2 shows that some models trained on only the source domain dataset generally perform better than the models also trained on the target domain. This puts the effectiveness of the proposed DG techniques strongly into question. Even the authors own model seems to perform generally better (also on the target domain!) when trained just on the source domain than when their DG scheme is used. Also, a model trained on a joint TD and SD dataset would be an interesting point of comparison.
  • When comparing the results to that of just using VS-GAT and training it on the source domain, the results from the proposed approach are significantly worse on the source domain, while also being rather similar in the target domain. 
  • The results of the ablation studies do not paint a clear picture towards the real benefit of the proposed changes to the IL paradigm, E-CBS, and T-Norm. In the unsupervised setting, E-CBS shows benefit in accuracy and recall but not in mAP in the target domain, but when employed together with T-Norm, the results are both in the source and target domains, except for recall in source domain. In the incremental setting, employing E-CBS helps on the source domain to improve the accuracy, and mAP in the target domain, but reduces the performance in other metrics. When employed together with T-Norm, it achieves the best mAP and Recall on the source domain, while resulting in worse performance in target domain. These inconsistencies and the lack of a clear goal / metric makes it hard to compare the advantages and disadvantages of the different techniques.
  • While the authors refer to it as T-Norm as present as a contribution, dividing the network output by a constant is a fairly frequent technique, e.g. softmax with temperature in the knowledge-distillation. Therefore, this would be considered an implementation detail rather than a contribution.
  •  E-CBS: The proposed "enhanced curriculum by smoothing" introduces Laplacian of Gaussian Kernels in addition to previously used Gaussian kernels. The difference or benefit of choosing the LoG kernel over the Gaussian kernel are however not reasoned about. The results shown in Table 5 show that this choice and positioning of kernels have varying impacts on performance, but the authors do not provide any explanation or intuition for these results. The proposal of E-CBS is therefore not entirely convincing.
  • T-Norm: The authors do not clearly describe the difference of their proposed T-Norm to conventional temperature normalization. It is therefore not clear how novel or meaningful their own contribution to this component is. Additionally, ablation results (e.g. Table 3, 4) do not show a clear indication of a performance improvement gained by employing T-Norm.
  • Non-public dataset: This work reports its main results on a non-public dataset, limiting reproducibility.

Overall, the paper provides improvement on existing techniques, and employs an interesting approach. The main weakness is the lack of a clear and objective goal, making it hard to compare the different methods, especially as different combinations are performing better and worse depending on the setting, and metric. With the current results, it is hard to judge if using E-CBS and T-Norm are beneficial ideas. 

 The authors address an important issue and provide an interesting approach. The individual components of this approach (IL, KD, Fine-Tuning, E-CBS, T-Norm) are studied in ablations. However, these ablations reveal that the performance improvements gained by each component seem small and conditional. When combined, the DG approach fails to demonstrate a clear general performance advantage even over models trained only on the source domain. The proposed approach is therefore unconvincing as a feasible method for Domain Generalization.

I suggest authors to clearly formulate the goal of their research, in terms of metrics. Assuming training only on the source domain leads to the best results on the source domain, and training only on the target domain leads to the best results on the target domain, one such metric could be to see how much the incrementally generalized model differs from these results. Using this metric, different techniques could be compared objectively. This could be used to better argue for the advantages of E-CBS and T-Norm. 

Additionally, the comparison with the VS-GAT, trained normally on the source domain shows mixed results. Picking a clear and objective metric would also help to argue why this proposed method is more beneficial.

Finally, I encourage the authors to publicly release their dataset to the community, if it is possible.

Minor comment:

The authors write: ‘As shown in Fig. 1, the visual features (Fvi)’. However, Fvi is not visible in figure 1 where F’’ and F’ are used. Please make the naming consistent

Author Response

We thank the reviewer for identifying and acknowledging the strengths of our paper. It is encouraging and helps keep our morals high. We also thank the reviewer for constructive feedbacks which has motivated us to strive for better.

Comment 1:

It is not clear if the main goal is to keep the performance on the original source domain or adapt to the target domain. This makes comparing different approaches challenging.

Response:

We thank the reviewer for highlighting this concern. The main goal of the paper is to incrementally domain generalize a graph network to the target domain without adversely scarifying its performance in the source domain. Additionally, we also present an improvement in graph network performance through the inclusion of T-Norm and our proposed E-CBS.  This paper presents two contributions (i) Incremental domain generalization. (ii) A minor improvement in the graph network performance (improvement is not links to domain generalization, just graph network) through the inclusion of T-Norm and our proposed E-CBS. Line 75-103 and be carefully reviewed and updated to highlight our goal and key contributions.

Comment 2:

Target Domain Performance: Comparing results from Table 1 and Table 2 shows that some models trained on only the source domain dataset generally perform better than the models also trained on the target domain. This puts the effectiveness of the proposed DG techniques strongly into question. Even the authors own model seems to perform generally better (also on the target domain!) when trained just on the source domain than when their DG scheme is used. Also, a model trained on a joint TD and SD dataset would be an interesting point of comparison.

Response:

We thank the reviewer for raising this clarification. When compared solely based on the accuracy (ACC), indeed, our model trained only on the source domain has better performance on the target domain. However, this can be attributed to the longtail issue since “idle” action is significantly higher than other actions in the detected interactions. To compare the model’s performance, mean average precision (mAP) and recall should also be considered. They help define the quality of positive predictions. While seen as a whole (mAP, ACC and recall), our model trained using incremental domain generalization technique offers better-balanced results. Additional justification is also provided in the results section, para 1, lines 226-234 (highlighted in red).

Comment 3:

When comparing the results to that of just using VS-GAT and training it on the source domain, the results from the proposed approach are significantly worse on the source domain, while also being rather similar in the target domain.

Response:

We thank the reviewer for raising this clarification. As stated in response to comment 2, the performance of the model must be seen as a whole (ACC, mAP and recall) instead of just based on ACC. Furthermore, the objective of our proposed technique is to domain generalize the model to the target domain without adversely scarifying its performance on the source domain (as stated in lines 83-86).

Comment 4:

The results of the ablation studies do not paint a clear picture of the real benefit of the proposed changes to the IL paradigm, E-CBS, and T-Norm. In the unsupervised setting, E-CBS shows benefit in accuracy and recall but not in mAP in the target domain, but when employed together with T-Norm, the results are both in the source and target domains, except for recall in the source domain. In the incremental setting, employing E-CBS helps on the source domain to improve the accuracy, and mAP in the target domain, but reduces the performance in other metrics. When employed together with T-Norm, it achieves the best mAP and Recall on the source domain, while resulting in worse performance in the target domain. These inconsistencies and the lack of a clear goal/metric make it hard to compare the advantages and disadvantages of the different techniques.

Response:

As stated in response to comment 1, the main objective of the E-CBS and T-Norm is to improve the graph’s network performance and not to improve incremental domain adaptation. Thus, their contribution to model performance can be viewed (i) only in the source domain for unsupervised domain generalization and (ii) in both the source and target domain for incremental learning. E-CBS was introduced to improve the model’s performance in terms of ACC, mAP and recall. T-Norm was adopted to study its effect on model calibration. When observing T-Norm’s contribution, it should be viewed in Fig. 3 (reliability diagram) instead of Table 2 since ACC, mAP and RECALL don’t give a clear picture on model calibration. Although incorporating T-Norm results in lower ACC, mAP and RECALL, as shown in Fig. 3, their predictions are more reliable. In the medical domain, both accuracy and reliability are important and must be balanced.

Comment 5:

While the authors refer to it as T-Norm as present as a contribution, dividing the network output by a constant is a fairly frequent technique, e.g. softmax with temperature in the knowledge-distillation. Therefore, this would be considered an implementation detail rather than a contribution.

Response:

We thank the author for raising this clarification. T-Norm is a frequently employed technique in knowledge distillation. The study on T-Norm’s effect in model calibration is claimed as one of the contributions, not its implementation.

Comment 6:

E-CBS: The proposed "enhanced curriculum by smoothing" introduces Laplacian of Gaussian Kernels in addition to previously used Gaussian kernels. The difference or benefit of choosing the LoG kernel over the Gaussian kernel is however not reasoned about. The results shown in Table 5 show that this choice and positioning of kernels have varying impacts on performance, but the authors do not provide any explanation or intuition for these results. The proposal of E-CBS is therefore not entirely convincing.

Response:

WE thank the reviewer for highlighting this. CBS employs Gaussian kernels to blur and limit the amount of features that propagates through the model and allow the model to learn progressively. As LoG gives attention to edges, replacing gaussian at every stage (different scale – image size reduces as it passes through different Conv layers), edges detected at initial stages (higher scale) could be dropped as the scale reduces. This could result in the loss of key features. Enforcing edge attention at multiple scales could have also induced noise. Having an LoG kernel at the initial layer to progressively reduce attention to edges and then using Gaussian kernels to progressively allow those edge features and surface features to pass through the network seems to offer a better solution.  These explanations are now added to the manuscript (lines 270-279).

Comment 7:

T-Norm: The authors do not clearly describe the difference of their proposed T-Norm to conventional temperature normalization. It is therefore not clear how novel or meaningful their own contribution to this component is. Additionally, ablation results (e.g. Table 3, 4) do not show a clear indication of a performance improvement gained by employing T-Norm.

Response:

There is no difference in the implementation of T-Norm (Temperature Normalization) and conventional temperature normalization. As stated in response to comment 5, the study on its effect on model calibration is claimed as one of our contributions. Its contribution to model performance is not shown in Tables 3 & 4 as its contribution to model calibration is reported using the reliability diagram (Fig. 3).

Comment 8:

Non-public dataset: This work reports its main results on a non-public dataset, limiting reproducibility.

Response:

We thank the reviewer for highlighting this. The source domain data has been made public through our past works (https://github.com/mobarakol/Surgical_SceneGraph_Generation). Upon acceptance of this paper, the target domain dataset will be made public using our GitHub repo.

Minor Comment 1:

The authors write: ‘As shown in Fig. 1, the visual features (Fvi)’. However, Fvi is not visible in figure 1 where F’’ and F’ are used. Please make the naming consistent.

Response:

We thank the reviewer for highlighting this. The manuscript has been updated (line 125).

Reviewer 2 Report

This paper focuses on the issue of domain shifts and the inclusion of new instruments and tissues. It is true that the learning domain generalization (DG) plays a pivotal role in expanding the instrument- tissue interaction detection to new domains in robotic surgery. It attempts to incorporate incremental DG on scene graphs to predict instrument-tissue interaction during robot-assisted surgery.  Quantitative and qualitative analysis proves that our incrementally- domain generalized interaction detection model was able to adapt to the target domain (transoral robotics surgery) while retaining its performance in the source domain (nephrectomy surgery).This research is interesting for the medical robot control research society. However, this paper has several limitations and the standard is not enough, and address the following items would result in a good paper,

  1. The literature review is not thorough about the application and the contributions. To highlight the contributions, it suggests reorganizing the section of the related work with real applications. It is recommended to read the following works and consider discussing their application scenarios in the introduction and discussion, for example, Towards Teaching by Demonstration for Robot-Assisted Minimally Invasive Surgery; Multi-Sensor Guided Hand Gesture Recognition for a Teleoperated Robot Using a Recurrent Neural Network
  2. The contribution of this paper is not clear. It suggests revising the contributions section and making these points clear and strong.
  3. The quality of the Figures should be improved and readable for the readers.
  4. Maybe it is better to discuss the possibility to improve the scope using deep learning to learn and optimize for online estimation in the introduction, for example, A Multimodal Wearable System for Continuous and Real-time Breathing Pattern Monitoring During Daily Activity
  5. It is recommended to present in the first section so that it can highlight the specific scope of this article. The meaning of the assessment experiment should be highlighted.
  6. There should be a further discussion about the limitation of the current works, in particular, what could be the challenge for its related applications. To let readers better understand future work, please give specific research directions.

Author Response

We thank the reviewer for their valuable comments.

Comment 1:

The literature review is not thorough about the application and the contributions. To highlight the contributions, it suggests reorganizing the section of the related work with real applications. It is recommended to read the following works and consider discussing their application scenarios in the introduction and discussion, for example, Towards Teaching by Demonstration for Robot-Assisted Minimally Invasive Surgery; Multi-Sensor Guided Hand Gesture Recognition for a Teleoperated Robot Using a Recurrent Neural Network.

Response:

We thank the reviewer for their valuable suggestions. We have added additional applications in section 1, para 1, lines 29-38 (highlighted in red).

Comment 2:

The contribution of this paper is not clear. It suggests revising the contributions section and making these points clear and strong.

Response:

We thank the reviewer for raising this concern. This paper has two contributions. Firstly, incremental domain generalization and secondly, a minor improvement in graph network (improvement not linked to domain generalization) through the inclusion of T-Norm and our proposed E-CBS. Line 75-103 and be carefully reviewed and updated to highlight our key contributions.

Comment 3:

The quality of the Figures should be improved and readable for the readers.

Response:

All three figures’ quality and readability have been improved.

Figure 1: Color and alignment have been updated to improve quality and readability.

Figure 2: Color and font have been updated to improve quality and readability.

Figure 3: Font updated to improve readability.

Comment 4 and 5:

Maybe it is better to discuss the possibility to improve the scope using deep learning to learn and optimize for online estimation in the introduction, for example, A Multimodal Wearable System for Continuous and Real-time Breathing Pattern Monitoring During Daily Activity. It is recommended to present in the first section so that it can highlight the specific scope of this article. The meaning of the assessment experiment should be highlighted.

Response:

We thank the reviewer for this suggestion. While our interaction detection model may not be suitable (interaction detection based on visual features) for applications related to monitoring breathing patterns during daily activities, the technique of domain generalization is generic to the deep learning domain and could potentially be used for such applications. However, as advised, we have added potential applications in the introduction section, para 1, lines 29-38 (highlighted in red).

Comment 6:

There should be a further discussion about the limitation of the current works, in particular, what could be the challenge for its related applications. To let readers better understand future work, please give specific research directions.

Response:

We thank the reviewer for this suggestion. We have added the current limitation and a possible direction for future works in the conclusion section lines 296-300 (highlighted in red).

Round 2

Reviewer 2 Report

Very good work and good response. The authors have addressed all of my concerns. The current version can be accepted. No more revision is required from my side.